# Integrated analyses of growth differentiation factor-15 concentration and cardiometabolic diseases in humans

Susanna Lemmelä[1]*[†], Eleanor M Wigmore[2]*[†], Christian Benner[1], Aki S Havulinna[1,3], Rachel MY Ong[4], Tibor Kempf[5], Kai C Wollert[5], Stefan Blankenberg[6,7,8], Tanja Zeller[6,8,9], James E Peters[10,11], Veikko Salomaa[3], Maria Fritsch[12], Ruth March[13], Aarno Palotie[1,14,15], Mark Daly[1,14,15], Adam S Butterworth[4,11,16,17], Mervi Kinnunen[1], Dirk S Paul[2,4,16], Athena Matakidou[2]

[1]Institute for Molecular Medicine Finland, University of Helsinki, Helsinki, Finland; [2]Centre for Genomics Research, AstraZeneca, Cambridge, United Kingdom; [3]Finnish Institute for Health and Welfare, Helsinki, Finland; [4]British Heart Foundation Cardiovascular Epidemiology Unit, Department of Public Health and Primary Care, University of Cambridge, Cambridge, United Kingdom; [5]Department of Cardiology and Angiology, Hannover Medical School, Hannover, Germany; [6]Clinic of Cardiology, University Heart and Vascular Center, University Medical Center Hamburg-Eppendorf, Hamburg, Germany; [7]Population Health Research Department, University Heart and Vascular Center, University Medical Center Hamburg-Eppendorf, Hamburg, Germany; [8]German Center for Cardiovascular Research (DZHK), partner site Hamburg/Kiel/Luebeck, Hamburg, Germany; [9]University Center of Cardiovascular Science, University Medical Center Hamburg-Eppendorf, Hamburg, Germany; [10]Department of Immunology and Inflammation, Imperial College London, London, United Kingdom; [11]Health Data Research UK Cambridge, Wellcome Genome Campus and University of Cambridge, Cambridge, United Kingdom; [12]Bioscience Renal, Research and Early Development Cardiovascular, Renal and Metabolism (CVRM), BioPharmaceuticals R&D, AstraZeneca, Gothenburg, Sweden; [13]Precision Medicine, Oncology R&D, AstraZeneca, Cambridge, United Kingdom; [14]Analytic and Translational Genetics Unit, Department of Medicine, Massachusetts General Hospital, Boston, United States; [15]Program in Medical and Population Genetics, Broad Institute of MIT and Harvard, Cambridge, United States; [16]British Heart Foundation Centre of Research Excellence, University of Cambridge, Cambridge, United Kingdom; [17]National Institute for Health Research Blood and Transplant Research Unit in Donor Health and Genomics, University of Cambridge, Cambridge, United Kingdom

*For correspondence:
susanna.lemmela@helsinki.fi (SL);
eleanor.wigmore@astrazeneca.com (EMW)

[†]These authors contributed equally to this work

**Abstract** Growth differentiation factor-15 (GDF15) is a stress response cytokine that is elevated in several cardiometabolic diseases and has attracted interest as a potential therapeutic target. To further explore the association of GDF15 with human disease, we conducted a broad study into the phenotypic and genetic correlates of GDF15 concentration in up to 14,099 individuals. Assessment of 772 traits across 6610 participants in FINRISK identified associations of GDF15 concentration with a range of phenotypes including all-cause mortality, cardiometabolic disease, respiratory diseases and psychiatric disorders, as well as inflammatory markers. A meta-analysis of genome-wide association studies (GWAS) of GDF15 concentration across three different assay platforms (n=14,099)

confirmed significant heterogeneity due to a common missense variant (rs1058587; p.H202D) in *GDF15*, potentially due to epitope-binding artefacts. After conditioning on rs1058587, statistical fine mapping identified four independent putative causal signals at the locus. Mendelian randomisation (MR) analysis found evidence of a causal relationship between GDF15 concentration and high-density lipoprotein (HDL) but not body mass index (BMI). Using reverse MR, we identified a potential causal association of BMI on GDF15 (IVW $p_{FDR}$ = 0.0040). Taken together, our data derived from human population cohorts do not support a role for moderately elevated GDF15 concentrations as a causal factor in human cardiometabolic disease but support its role as a biomarker of metabolic stress.

## Editor's evaluation

This integrated observational and genetic analysis using data from large biobanks comprehensively investigated the role of Growth Differentiation Factor-15 in a wide range of human diseases and will be of interest to cardiometabolic disorder researchers. GDF-15 appears to be a marker of metabolic stress rather than having a causative role.

## Introduction

Obesity accounts for an estimated 2.8 million deaths worldwide and 2.3% of global disability-adjusted life years (*World Health Organization, 2008*). Obesity has been causally linked to a variety of cardiometabolic risk factors and diseases, including fasting insulin, systolic blood pressure, and type 2 diabetes (*Holmes et al., 2014*). Therefore, interventions to reduce obesity are recommended in the management of these diseases (*Yumuk and Tsigos, 2016*). Currently, the most effective obesity intervention is bariatric surgery (*Jammah, 2015*); however, its clinical utility is limited as it is highly invasive and accompanied by a high risk of long-term complications such as gastroesophageal reflux disease and nutritional deficiencies (*Mesureur and Arvanitakis, 2017*). New therapies such as GLP-1 receptor agonists aim to target obesity via appetite suppression but are associated with common side effects including nausea and diarrhoea (*Wilding et al., 2021*). The development of additional therapeutic strategies targeting obesity as an underlying cause and pathology of cardiometabolic disease represents an area of unmet clinical need.

GDF15, a distant member of the transforming growth factor-β family (*Bootcov et al., 1997*; *Emmerson et al., 2018*), is upregulated in response to cellular stress and several diseases, including cancer. It has been suggested that GDF15 functions as a stress response agent implicated in organ injury (*Kempf et al., 2006*; *Tsai et al., 2018*; *Zimmers et al., 2005*). GDF15 was originally identified to play a role in tumour-induced anorexia/cachexia in mice through appetite regulation (*Johnen et al., 2007*). The body weight reduction is considered to be mainly driven by food intake inhibition mediated by its receptor GDNF-family receptor alpha like (GFRAL) in distinct areas of the brainstem (*Emmerson et al., 2017*; *Hsu et al., 2017*; *Mullican et al., 2017*; *Yang et al., 2017*). GDF15 plasma levels are elevated in animal models of obesity (*Lockhart et al., 2020*; *Xiong et al., 2017*); however, administration of recombinant protein robustly lowers body weight in obese and diabetic animals, including non-human primates (*Mullican et al., 2017*; *Xiong et al., 2017*). Similarly, genetic overexpression of Gdf15 results in decreased body weight and increased resistance to obesity associated with high-fat diets, whereas Gdf15 and Gfral deficient mice are more susceptible to diet-induced obesity (*Mullican et al., 2017*; *O'Rahilly, 2017*; *Tran et al., 2018*). Recent preclinical work has identified GDF15's role as a sentinel protein, upregulated in response to various ingested toxins triggering nausea-related behaviour in rodents assessed by pica and CTA (Conditioned Taste Aversion: *Borner et al., 2020b*; *Patel et al., 2019*). It is hypothesised that this contradictory finding of upregulated levels in obesity but weight loss upon overexpression may be explained by GDF15 acting as a signal to the brain to induce weight loss (*Villanueva, 2017*).

Currently, there is a lack of data supporting GDF15 as a potential therapeutic target in humans. However, observational studies have revealed strong positive correlations of GDF15 plasma levels with body mass index (BMI), insulin resistance, age, and mean arterial blood pressure (in obese individuals: *Tsai et al., 2015*; *Vila et al., 2011*). Higher GDF15 levels have also been associated with all-cause mortality as well as mortality associated with heart failure and acute myocardial infarction, cancer, advanced heart failure, and end-stage chronic kidney disease (*Adela and Banerjee, 2015*; *Khan et al.,*

*2009*; *Nair et al., 2017*; *Wiklund et al., 2010*). Metformin therapy, a type 2 diabetes treatment, has been shown to depend on GDF15 to lower body weight in mice and plasma GDF15 levels increased up to 40% upon metformin therapy in humans (*Coll et al., 2020*; *Day et al., 2019*). In fact, an analysis of changes induced by Metformin treatment across 237 biomarkers showed GDF15 levels to be most significantly altered, independent of glucose and glycosylated haemoglobin (*Gerstein et al., 2017*). Recent data also suggest that GDF15 is involved in hyperemesis gravidarum (*Fejzo et al., 2018a*), supporting the hypothesis that GDF15 triggers anorexia and subsequent weight loss, at least partly, through the induction of malaise (*Borner et al., 2020a*). Mendelian randomisation (MR) has previously been applied to explore the causal relationship of GDF15 levels and cardiometabolic diseases, with causal associations reported for high-density lipoprotein (HDL) cholesterol and bone mineral density (BMD: *Cheung et al., 2019*; *Folkersen et al., 2020*). However, these analyses were based on small genetic studies for GDF15 and have not been replicated. Opposing results have also been reported regarding the association of GDF15 on BMI, with one study finding a causal relationship (*Karhunen et al., 2021*) and another reporting no significant effect (*Au Yeung et al., 2019*).

In this study, we used data from several large biobanks to conduct a systematic and extensive phenotypic and genotypic analysis of GDF15 with cardiometabolic traits and diseases, and to ascertain the causal relationship between GDF15 levels and cardiometabolic traits using MR and protein-truncating variant (PTV) analysis.

# Results
## Association of GDF15 plasma levels with 676 disease outcomes

To assess systematically the relationship between GDF15 plasma levels and clinical phenotypes, we utilised a large Finnish biobank, FINRISK. The FINRISK cohort comprises a cross-sectional population survey carried out over a 40-year period in Finland. GDF15 plasma levels were available for 6610

**Table 1.** Disease endpoints associated ($p_{FDR} < 1 \times 10^{-5}$) with GDF15 plasma levels in FINRISK.

| Disease endpoint | Cases/controls | OR (95% CI) | $p_{FDR}$ |
|---|---|---|---|
| All-cause mortality | 1057/5481 | 1.79 (1.68–1.90) | $7.5 \times 10^{-24}$ |
| Death due to cardiac causes | 471/6067 | 1.76 (1.61–1.90) | $3.0 \times 10^{-11}$ |
| Atherosclerosis, excluding cerebral, coronary, and PAD | 379/6159 | 1.67 (1.51–1.82) | $3.2 \times 10^{-8}$ |
| Diabetes mellitus type 2 | 567/5971 | 1.48 (1.36–1.60) | $3.2 \times 10^{-8}$ |
| Diabetes mellitus | 592/5946 | 1.44 (1.33–1.56) | $9.8 \times 10^{-8}$ |
| Diseases of arteries, arterioles, and capillaries | 505/6033 | 1.48 (1.35–1.61) | $3.6 \times 10^{-7}$ |
| Other COPD | 221/6317 | 1.84 (1.63–2.04) | $5.7 \times 10^{-7}$ |
| COPD | 235/6303 | 1.77 (1.57–1.97) | $9.6 \times 10^{-7}$ |
| Pneumonia (excl. viral and due to other infectious organisms not elsewhere classified) | 779/5759 | 1.34 (1.24–1.44) | $9.6 \times 10^{-7}$ |
| Type 2 diabetes without complications | 485/6053 | 1.44 (1.31–1.57) | $1.2 \times 10^{-6}$ |
| All pneumonia | 789/5749 | 1.33 (1.23–1.43) | $1.9 \times 10^{-6}$ |
| Chronic kidney disease | 66/6472 | 2.46 (2.14–2.79) | $4.4 \times 10^{-6}$ |
| Influenza and pneumonia | 833/5795 | 1.30 (1.21–1.40) | $4.4 \times 10^{-6}$ |
| Type 2 diabetes with renal complications | 35/6503 | 2.97 (2.57–3.38) | $8.1 \times 10^{-6}$ |
| Sequelae of cerebrovascular disease | 209/6329 | 1.67 (1.47–1.86) | $9.2 \times 10^{-6}$ |
| Alcoholic liver disease | 56/6482 | 2.26 (1.95–2.57) | $9.2 \times 10^{-6}$ |

Results are adjusted for age, gender, smoking, and BMI. Abbreviations: OR, odds ratio; CI, confidence interval; COPD, chronic obstructive pulmonary disease; PAD, peripheral artery disease; SAH, subarachnoid haemorrhage; GDF15, growth differentiation factor-15. ICD codes for these disease endpoints have been published (*Tuomo et al., 2020*).

participants from the 1997 recruitment cohort linked to 676 disease outcomes and were included in this analysis.

The median GDF15 concentration (measured using an immunoluminometric assay [ILMA]) was 796 ng/L (interquartile range [IQR] = 664–986). First, we examined the association of baseline characteristics with GDF15 plasma levels (*Supplementary file 1a*). Age explained 28% of the observed GDF15 variance, with current smoking and BMI accounting for 1.9% and 0.08% of the variance, respectively (*Supplementary file 1b*). Gender did not show a significant association. All subsequent analyses have been corrected for these covariates (age, gender, smoking, and BMI). Analyses corrected for age and gender alone are also presented for comparison (*Supplementary file 2a-b*), although results were similar.

We then investigated potential associations of GDF15 plasma levels with a range of disease phenotypes (both prevalent and incident), focusing on the phenotypes defined by the FinnGen consortium (*Tuomo et al., 2020*), as these endpoints have undergone additional clinical validation (see Materials and methods). A total of 676 disease endpoints were examined as dependent variables in association analyses with GDF15 plasma levels (the independent variable). After multiple-testing correction using false discovery rate (FDR, $p_{FDR} < 0.05$), GDF15 was significantly associated with 80 disease endpoints (*Table 1* and *Supplementary file 2c*).

The most significant clinical associations observed were with all-cause mortality (logistic regression odds ratio, OR = 1.8, CI = 1.7–1.9, p-value = $9.8 \times 10^{-27}$) and mortality due to cardiac diseases (OR = 1.8, CI = 1.6–1.9, p-value = $7.7 \times 10^{-14}$). We also observed significant associations between GDF15 levels and type 2 diabetes (OR = 1.5, CI = 1.3–1.6, p-value = $2.2 \times 10^{-9}$). Further, GDF15 plasma levels were significantly associated with cardiovascular diseases (OR = 1.2, CI = 1.1–1.2, p-value = $5.6 \times 10^{-6}$), as well as subtype endpoints such as atherosclerosis, hypertension, peripheral artery (PAD) disease and stroke, and chronic kidney disease (OR = 2.46, CI = 2.1–2.8, p-value = $4.4 \times 10^{-6}$: *Table 1*). Additional positive associations were observed with respiratory diseases, such as chronic obstructive pulmonary disease (COPD) and pneumonia, and psychiatric disorders, including schizophrenia and mental and behavioural disorders due to psychoactive substance use. Cancer phenotypes were also associated with GDF15 levels, including malignant neoplasm of the respiratory system and intrathoracic organs (OR = 1.6, CI = 1.3–1.8, p-value = 0.0015) and malignant neoplasm of the bronchus and lung (OR = 1.6, CI = 1.3–1.9, p-value = 0.0029).

## GDF15 plasma concentration associates with prevalent and incident disease and independently predicts all-cause mortality and cardiometabolic diseases

To determine the prognostic potential of GDF15 in predicting incident disease, we investigated its plasma levels in relation to prevalent and incident diseases in the FINRISK cohort. We selected disease endpoints showing association to the GDF15 plasma levels, namely type 2 diabetes, atherosclerosis (excluding cerebral, coronary, and PAD), COPD, psychiatric disorders, and malignant neoplasm of respiratory system and intrathoracic organs (*Supplementary file 2d*).

Type 2 diabetics (prevalent cases) had a 1.3-fold increase in median GDF15 levels compared to non-diabetic controls (median difference = 222 ng/L). Furthermore, GDF15 levels were significantly higher in prevalent diabetics (diagnosed at or before plasma sampling, n=37; median = 1204 ng/L) compared to both incident cases (diagnosed after plasma sampling, n=500; median = 992 ng/L; Wilcoxon rank sum test p-value = 0.008) and non-diabetic controls (n=6003; 784 ng/L; Wilcoxon rank sum test p-value = $5.2 \times 10^{-11}$). GDF15 plasma levels were also significantly higher in incident cases than in non-diabetic controls (Wilcoxon rank sum test p-value = $6.2 \times 10^{-51}$).

Similarly, GDF15 levels were significantly higher in prevalent atherosclerosis and prevalent COPD cases (n=34, median = 1224 ng/L and n=30, 1356 ng/L, respectively) compared to respective incident cases (n=357, median = 1104 ng/L, Wilcoxon rank sum test p-value = 0.02 and n=244, median = 1119 ng/L; Wilcoxon rank sum test p-value = 0.02, respectively) and healthy controls (n=6219, 784 ng/L, Wilcoxon rank sum test p-value = $4.69 \times 10^{-13}$ and n=6366; 789 ng/L, Wilcoxon rank sum test p-value = $3.10 \times 10^{-11}$, respectively). GDF15 plasma levels were also higher in incident cases than healthy controls (Wilcoxon rank sum test p-value = $9.6 \times 10^{-78}$ and Wilcoxon rank sum test p-value = $1.6 \times 10^{-51}$, respectively).

Other tested endpoints (psychiatric disorders; malignant neoplasm of respiratory system and intrathoracic organs) also showed similar results for which plasma GDF15 levels seemed to predict the disease (*Supplementary file 2d*). Together, these results of several disease endpoints indicate that increased GDF15 plasma levels could represent an early biomarker in individuals before disease diagnosis.

Next, we carried out a Cox regression survival analysis on all-cause mortality, type 2 diabetes, and cardiovascular disease. During a 10-year follow-up period, 393 (6%) study subjects had died, 97 (2%) developed diabetes, and 438 (7%) developed cardiovascular disease. Models accounted for blood pressure medication, smoking, total cholesterol, HDL cholesterol, BMI, prevalent cardiovascular disease, and mean systolic blood pressure. Our results revealed that GDF15 plasma concentration is an independent predictor of all-cause mortality (hazard ratio, HR = 1.7, p-value = $9.2 \times 10^{-12}$), type 2 diabetes (HR = 1.40, p-value = 0.02), and cardiovascular disease (HR = 1.4, p-value = $1.5 \times 10^{-6}$, *Figure 1*, *Figure 1—figure supplement 1* and *Supplementary file 2e*). Individuals with GDF15 plasma levels above 967 ng/L (the upper quartile) were more than twice as likely to develop type 2 diabetes compared to individuals with lower levels (HR = 2.2, CI = 1.4–3.5, p-value = $7.1 \times 10^{-4}$).

## Association of GDF15 plasma levels with 96 quantitative biomarkers

We then performed an association analysis of GDF15 plasma levels with 96 quantitative biomarkers in FINRISK. After multiple-testing correction ($p_{FDR} < 0.05$), significant associations were observed with 45 biomarkers, most notably, with blood markers known to be associated with inflammation (*Supplementary file 2f*).

The most significantly associated biomarkers were mid-regional pro-adrenomedullin (MR-proADM, beta = 0.24, p-value = $1.2 \times 10^{-91}$), C-reactive protein (CRP, beta = 0.22, p-value = $4.6 \times 10^{-64}$) and hepatocyte growth factor (HGF, beta = 0.17, p-value = $4.2 \times 10^{-43}$). All three markers have been previously reported to be non-specifically elevated in a number of human diseases and have known prognostic value (*Jackson et al., 2016*; *Madonna et al., 2012*; *Matsumoto et al., 2017*; *Miller et al., 2007*; *Peacock, 2014*). Significant associations were also observed between GDF15 and components of metabolic syndrome, specifically, levels of serum triglycerides (beta = 0.13, p-value = $2.4 \times 10^{-22}$), fasting insulin (beta = 0.10, p-value = $1.2 \times 10^{-13}$) and waist-to-hip ratio (WHR) (beta = 0.049, p-value = $7.0 \times 10^{-10}$).

## Genome-wide association studies of plasma levels of GDF15 quantified using three different assays in two independent cohorts

We performed GWAS to identify the genetic determinants of GDF15 plasma levels using two independent cohorts (FINRISK and INTERVAL) and three GDF15 quantification methodologies (ILMA in FINRISK and Olink and SomaScan proteomic assays in INTERVAL). Results from individual GWAS analyses are reported in *Supplementary file 3a-c*. In FINRISK GWAS was conducted on 5817 individuals with available GDF15 plasma levels, measured using an ILMA (*Sinning et al., 2017*; *Supplementary file 3a*). The most significant associations were found within or in proximity to the *GDF15* gene on chromosome 19, comprising 159 significant variants (p-value < $5 \times 10^{-8}$). In INTERVAL we assayed plasma GDF15 in 3301 participants using SomaScan (an aptamer-based multiplex protein assays) and in 4998 participants using a proximity extension-based antibody assay (Olink). The most significant associations were found around the *GDF15* gene on chromosome 19, comprising 134 significant variants for INTERVAL-SomaScan and 72 significant variants for INTERVAL-Olink (p-value < $5 \times 10^{-8}$:*Supplementary file 3b-c*).

We then performed a meta-analysis of the genome-wide significant variants identified in the GWAS performed in the FINRISK, INTERVAL-SomaScan, and INTERVAL-Olink cohorts to generate a combined statistic and heterogeneity value. The analysis identified four genetic variants (rs1059369, rs1054221, rs1227734, rs189593084) at the *GDF15* locus to be associated with GDF15 plasma levels (p-value < $5 \times 10^{-8}$: *Supplementary file 3d*).

Two variants had a strengthened signal in the combined meta-analysis, rs1054221 and rs1227734 (*Supplementary file 3d*), However, five genetic associations, which were genome-wide significant for two of the three GWAS, displayed substantial heterogeneity, for example, rs16982345 (heterogeneity $I^2$=99.8%, heterogeneity p-value = $7.6 \times 10^{-180}$; *Supplementary file 3d*). By exploring the LD structure between the heterogeneous variants (*Supplementary file 3e*), we found that these variants were all

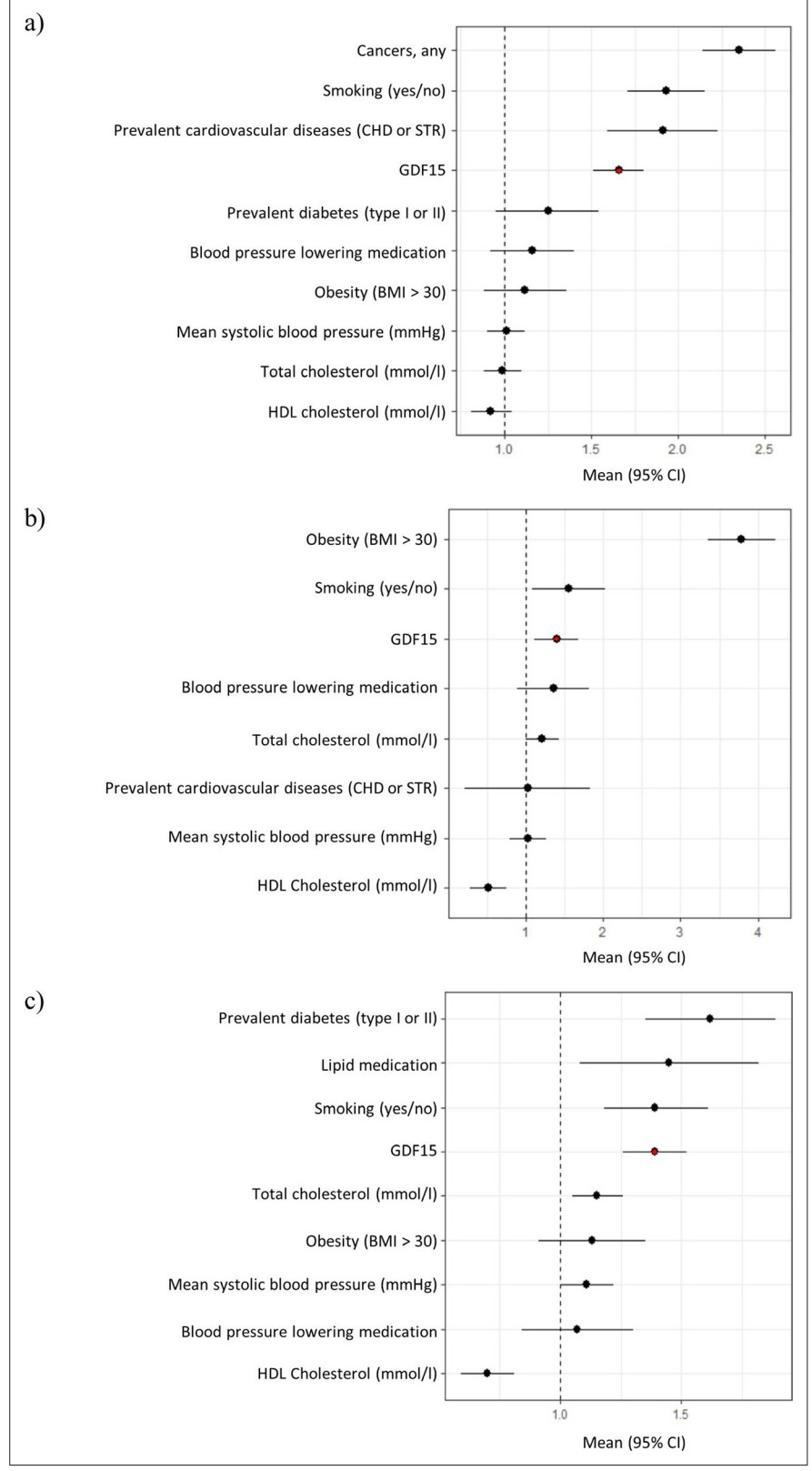

**Figure 1.** Forest plots of Cox proportional hazard models for independent predictors of (**a**) all-cause mortality, (**b**) diabetes, and (**c**) cardiovascular disease. The plot reports hazard ratios and 95% condidence intervals (error bars) with the dashed line representing the null effect. GDF15 is highlighted in red and variables are ordered by highest hazards ratio. Sample sizes are as follows; (a) n=393, (b) n=97 and (c) n=438. Abbreviations: BMI, body

*Figure 1 continued on next page*

*Figure 1 continued*

mass index; GDF15, growth differentiation factor-15; CHD, coronary heart disease; STR, stroke; HDL, high-density lipoprotein.

The online version of this article includes the following figure supplement(s) for figure 1:

**Figure supplement 1.** Survival curves of Cox proportional hazards model for (**a**) all-cause mortality (**b**) diabetes, and (**c**) cardiovascular disease stratified by GDF15 quartiles.

located within the same LD block that included the missense variant rs1058587 (p.H202D). Heterogeneity in GWAS results of GDF15 levels has been observed in other studies (*Jiang et al., 2018*; *Pietzner et al., 2020*) and significant variations in GDF15 antibody binding affinity dependent on rs1058587 have been previously reported (*Fairlie et al., 2001*).

## Mitigation of epitope binding effects due to the rs1058587 missense variant

To remove any potential epitope effects, we applied conditional analysis on rs1058587 when linking plasma GDF15 concentrations with disease phenotypes and quantitative traits. A total of 59 significant disease associations and 43 biomarker associations were identified. These associations largely did not differ from the unconditioned results, although 21 fewer disease associations reached significance (*Supplementary file 2g-h*).

To minimise confounding of the GWAS results from rs1058587 epitope effects, we conditioned the association signal on chromosome 19 on rs1058587 in both FINRISK (*Supplementary file 3f*) and INTERVAL-SomaScan (*Supplementary file 3g*). INTERVAL-Olink did not have a significant association at this LD block. We then performed a meta-analysis across the genome for all three cohorts. Only signals within 1 Mb of the GDF15 gene reached significance in the meta-analysis. Manhattan and Quantile-Quantile (QQ) plots of this GWAS meta-analysis are shown in *Table 2*, *Figure 2*, and *Supplementary file 3h*. We found 146 significant associations in this region, with rs1227734 identified as the strongest association (beta = 0.50, p-value = $2.5 \times 10^{-187}$).

Fine-mapping this conditioned meta-analysis revealed four association signals with a causality probability of 0.86. The top configuration consisted of variants rs1054221 (beta = −0.50, p-value = $2.4 \times 10^{-184}$), rs3787023 (beta = 0.04, p-value = 0.0017), rs138515339 (beta = −0.27, p-value = $3.2 \times 10^{-11}$), and rs141542836 (beta = 0.61, p-value = $1.9 \times 10^{-40}$) and had a regional heritability of 7%. Functional annotation can be found in *Supplementary file 3i*.

## MR analysis suggests a causal relationship between GDF15 plasma levels and HDL

Observed associations of GDF15 plasma levels with obesity-related diseases have been presented here and found in previous reports (*Tsai et al., 2015*; *Vila et al., 2011*) and variants in the *GDF15* gene region have been associated with cardiovascular traits, cholesterol, WHR, and BMI (*Wang et al., 2021a*). We therefore applied MR to assess the relationship between genetically determined GDF15 plasma levels and BMI, WHR, glucose, and type 2 diabetes. We also included HDL cholesterol and estimated BMD (eBMD) as additional outcomes in our MR analysis due to the positive MR results in previous studies (*Cheung et al., 2019*; *Folkersen et al., 2020*).

We applied two-sample MR (*Bowden et al., 2015*) using LD clumping ($R^2 < 0.01$, within 1.5 Mb of either side of the lead SNP, see Materials and methods) and identified five independent genetic instruments from the conditioned meta-analysis on chromosome 19: rs111527728, rs113700483, rs1227734, rs150286074, and rs73923175 (regional heritability = 0.061, F-statistics reported in *Supplementary file 4a*). Association statistics between the conditioned instrumental variables (IVs) and exposure (GDF15) were taken from the above meta-analysis, and the association statistics between the IVs and outcome were extracted from publicly available GWAS summary statistics (see Materials and methods). These values were applied to a random-effect inverse variance weighted (IVW) MR, as well as MR-Egger, weighted median MR, and MR-PRESSO methods (*Verbanck et al., 2018*), in order to detect horizontal pleiotropy (see Materials and methods for further details on the underlying assumptions of these methods). MR using the conditioned variants found two significant associations of genetically determined GDF15 plasma levels with HDL (IVW estimate = −0.0085, $p_{FDR}$

**Table 2.** Meta-analysis of GDF15 GWAS conditioned on rs1058587 in FINRISK and INTERVAL.

| SNPs | LD block | EA/OA | FINRISK | | INTERVAL-SomaScan | | INTERVAL-Olink | | Meta-analysis | | | |
|---|---|---|---|---|---|---|---|---|---|---|---|---|
| | | | beta | p-value | beta | p-value | beta | p-value | beta | p-value | Heterogeneity I² | Heterogeneity p-value |
| rs16982345 | 1 | A/G | -0.28 | 0.55 | 0.30 | 0.18 | 0.01 | 0.72 | 0.01 | 0.64 | 4.2 | 0.35 |
| rs1058587 | 1 | G/C | – | – | – | – | 0.01 | 0.77 | – | – | – | – |
| rs3787023 | 1 | A/G | 0.06 | 0.0011 | -0.011 | 0.72 | 0.04 | 0.072 | 0.04 | 0.0017 | 51.4 | 0.13 |
| rs1055150 | 1 | G/C | 0.06 | 0.0012 | -0.017 | 0.56 | 0.04 | 0.080 | 0.04 | 0.0025 | 58.5 | 0.090 |
| rs1059369 | 1 | A/T | 0.06 | 0.0010 | -0.018 | 0.54 | 0.04 | 0.078 | 0.04 | 0.0024 | 60.2 | 0.081 |
| rs1054221 | 2 | C/T | 0.38 | $3.4\times10^{-37}$ | 0.62 | $7.8\times10^{-83}$ | 0.52 | $9.4\times10^{-74}$ | 0.50 | $2.4\times10^{-186}$ | 93.4 | $2.9\times10^{-7}$ |
| rs1227734 | 2 | T/C | 0.38 | $3.9\times10^{-37}$ | 0.62 | $1.8\times10^{-83}$ | 0.51 | $7.1\times10^{-74}$ | 0.50 | $2.5\times10^{-187}$ | 93.1 | $5.4\times10^{-7}$ |
| rs189593084 | 3 | A/C | -0.33 | $8.4\times10^{-14}$ | -0.61 | 0.011 | -0.47 | 0.0053 | -0.35 | $1.3\times10^{-16}$ | 0.0 | 0.40 |

For comparison identical variants to those listed in **Supplementary file 3d** are shown here (these variants were identified by fine mapping unconditioned GWAS results from FINRISK and INTERVAL). LD blocks were defined as SNPs that had LD > 0.1 with the lead variant (most significantly associated variant). Abbreviations: GDF15, growth differentiation factor-15; GWAS, genome-wide association study.

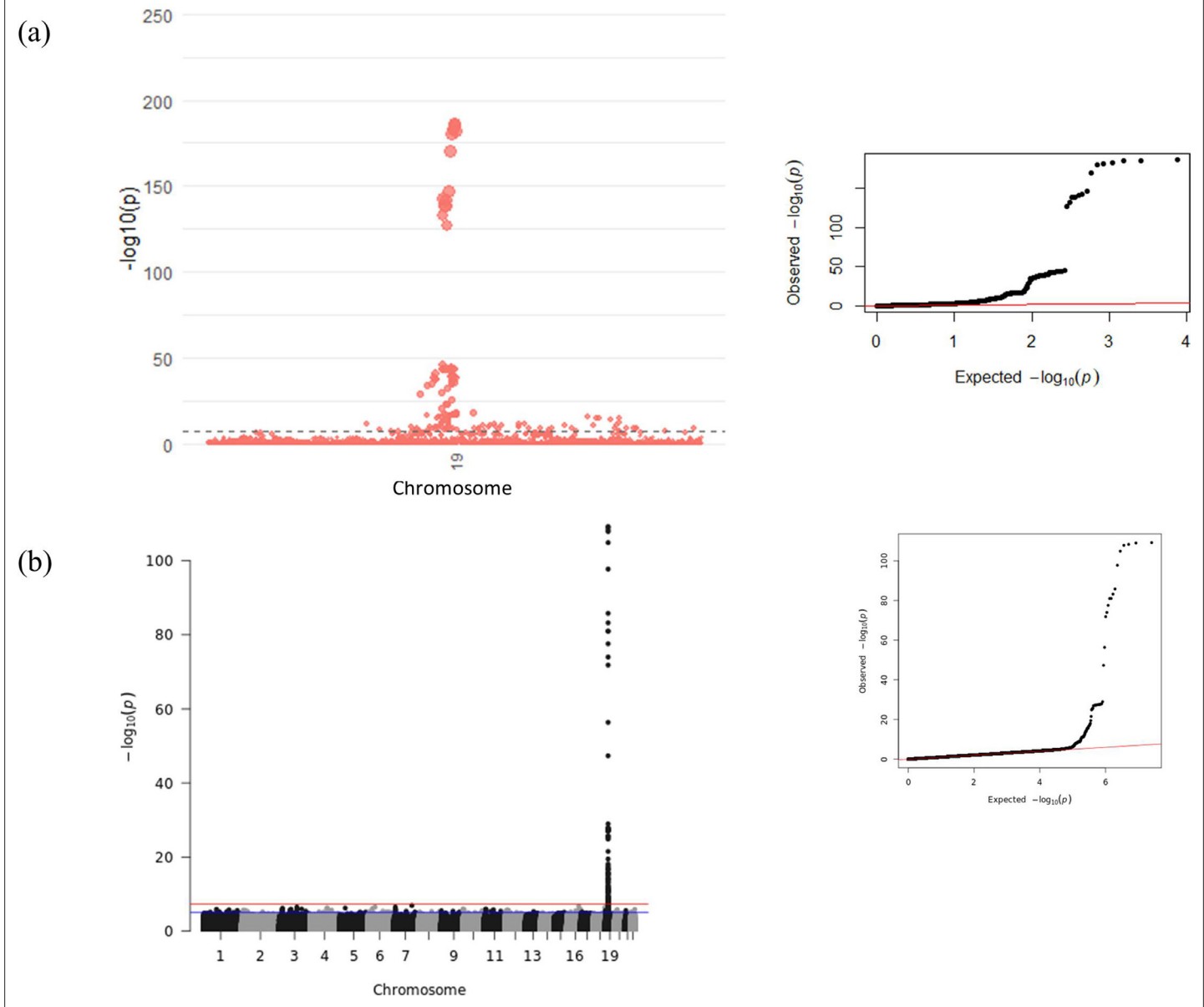

**Figure 2.** Manhattan and Quantile-Quantile (QQ) plots for genome-wide association study (GWAS) meta-analysis of conditioned growth differentiation factor-15 (GDF15) plasma levels in 14,099 individuals for (**a**) the *GDF15* region and (**b**) all chromosomes. The dotted line (**a**) and red line (**b**) represent genome-wide significance (p-value < 5 × 10⁻⁸).

= 0.0024) and WHR (IVW estimate = 0.017, $p_{FDR}$ = 0.0013). The effects sizes for both of these were very small and the WHR result was not robust to multiple-testing correction in MR-PRESSO (*Table 3* and *Supplementary file 4b*). Horizontal pleiotropy was not detected in any of the MR analyses. We did not replicate findings of causality of genetically determined GDF15 plasma levels with eBMD but we were able to provide additional validation of the causal role of genetically determined GDF15 in HDL cholesterol (*Cheung et al., 2019*; *Folkersen et al., 2020*).

## Reverse MR analysis identifies BMI as a causal factor for GDF15 plasma levels

We applied reverse two-sample MR (as described above) using GDF15 plasma levels as the outcome variable to assess associations with BMI, WHR, glucose, diabetes, HDL cholesterol, and eBMD as the MR exposures. GDF15 GWAS summary statistics were taken from the conditioned meta-analysis

**Table 3.** Mendelian randomisation results for genetically determined GDF15 plasma levels as the exposure with cardiometabolic outcomes.

| | SNPs | IVW (random) | | MR-Egger | | | MR-PRESSO | | | | | |
| | | Estimate (SE) | P_FDR (raw p-value) | Estimate (SE) | p-value | Intercept p-value | Estimate (SE) | P_FDR (raw p-value) | Outlier estimate | Outlier p-value | Global p-value | Distortion p-value |
|---|---|---|---|---|---|---|---|---|---|---|---|---|
| BMI | 5 | −0.0025 (0.012) | 0.90 (0.84) | −0.14 (0.023) | 0.53 | 0.53 | −0.0025 (0.012) | 0.89 (0.85) | – | – | 0.31 | – |
| WHR | 5 | **0.017 (0.0053)** | **0.0013 (0.0039)** | 0.0058 (0.0093) | 0.53 | 0.14 | 0.017 (0.0049) | 0.078 (0.026) | – | – | 0.50 | – |
| Diabetes | 5 | 0.014 (0.018) | 0.43 (0.86) | −0.20 (0.031) | 0.53 | 0.19 | 0.014 (0.018) | 0.89 (0.48) | – | – | 0.58 | – |
| Glucose | 5 | −0.00096 (0.0080) | 0.90 (0.90) | −0.0067 (0.014) | 0.63 | 0.62 | −0.00096 (0.0068) | 0.89 (0.89) | – | – | 0.83 | – |
| HDL | 5 | **−0.0085 (0.0023)** | **0.0024 (0.0014)** | −0.0069 (0.0041) | 0.092 | 0.62 | **−0.0085 (0.0013)** | **0.019 (0.0031)** | – | – | 0.79 | – |
| eBMD | 5 | 0.0047 (0.012) | 0.69 (0.90) | −0.0072 (0.022) | 0.75 | 0.51 | 0.0047 (0.012) | 0.89 (0.71) | – | – | 0.28 | – |

A significant finding of pleiotropy is indicated by the intercept p-value in MR-Egger and the Global p-value in MR-PRESSO. For MR-PRESSO the outlier test is only run if an outlier is detected. The distortion p-value represents whether the outlier removal significantly reduces the horizontal pleiotropy. Significant findings p-value < 0.05 are marked in bold text.
Abbreviations: GDF15, growth differentiation factor-15; IVW, inverse variance weighted; BMI, body mass index; WHR, waist-hip ratio; HDL, high-density lipoprotein; eBMD, estimated bone mineral density.

**Table 4.** Reverse Mendelian randomisation with GDF15 plasma levels as outcome.

| | | IVW (random) | | MR-Egger | | | MR-PRESSO | | | | | |
|---|---|---|---|---|---|---|---|---|---|---|---|---|
| | SNPs | Estimate (SE) | P_FDR (raw p-value) | Estimate (SE) | p-value | Intercept p-value | Raw estimate (SE) | P_FDR (raw p-value) | Outlier estimate | Outlier p-value | Global p-value | Distortion p-value |
| BMI | 1011 | **0.097 (0.028)** | **0.0040 (0.00066)** | 0.14 (0.082) | 0.087 | 0.58 | 0.097 (0.028) | 0.0041 (0.00069) | – | – | 0.11 | – |
| WHR | 588 | 0.040 (0.039) | 0.45 (0.30) | −0.089 (0.11) | 0.41 | 0.21 | −0.040 (0.039) | 0.45 (0.30) | – | – | **0.00125** | – |
| Diabetes | 278 | 0.00082 (0.015) | 0.96 (0.96) | −0.055 (0.031) | 0.071 | **0.038** | −0.00082 (0.015) | 0.95 (0.95) | – | – | **0.00088** | – |
| Glucose | 77 | −0.065 (0.056) | 0.45 (0.25) | −0.24 (0.13) | 0.065 | 0.14 | 0.065 (0.056) | 0.45 (0.26) | – | – | 0.18 | – |
| HDL | 483 | −0.044 (0.061) | 0.56 (0.47) | −0.057 (0.096) | 0.56 | 0.87 | 0.044 (0.061) | 0.55 (0.46) | 0.054 (0.060) | 0.37 | **0.0147** | 0.89 |
| eBMD | 1113 | 0.030 (0.019) | 0.36 (0.12) | 0.032 (0.036) | 0.37 | 0.94 | −0.030 (0.019) | 0.36 (0.12) | −0.027 (0.018) | 0.14 | **<0.000125** | – |

A significant finding of pleiotropy is indicated by the intercept p-value in MR-Egger and the Global p-value in MR-PRESSO. For MR-PRESSO the outlier test is only run if an outlier is detected. The distortion p-value represents whether the outlier removal significantly reduces the horizontal pleiotropy. Significant findings p-value < 0.05 are marked in bold text.
Abbreviations: GDF15, growth differentiation factor-15; IVW, inverse variance weighted; BMI, body mass index; WHR, waist-hip ratio; HDL, high-density lipoprotein; eBMD, estimated bone mineral density.

of GDF15 levels in FINRISK and INTERVAL in all chromosomes. LD clumping was used to identify the genetic instruments (see Materials and methods, F-statistics are reported in *Supplementary file 4c-h*). We found a significant association between higher genetically predicted BMI and higher GDF15 plasma levels (IVW estimate = 0.097, $p_{FDR}$ = 0.0040) but not any other tested trait (*Table 4* and *Supplementary file 4i*). Sensitivity analyses in MR-PRESSO but not MR-Egger analysis confirmed the association with BMI (p-value > 0.05). No horizontal pleiotropy was detected in this association with MR-Egger or MR-PRESSO. Horizontal pleiotropy was identified in diabetes with MR-Egger and MR-PRESSO. MR-PRESSO additionally identified horizontal pleiotropy with WHR, HDL, and eBMD.

## Effect of GDF15 PTVs on cardiometabolic traits in 302,388 participants of UK Biobank

Carriers of *GDF15* PTVs present an opportunity to explore the phenotypic consequences of predicted loss-of-function (LOF) of GDF15 on human disease. We analysed whole-exome sequencing data from 302,388 participants from the UK Biobank, of whom 109 carried *GDF15* PTVs in the heterozygous state. We assessed differences in BMI, WHR, glucose, eBMD, and HDL cholesterol between carriers and non-carriers using the Mann-Whitney U test. We were not able to carry out this analysis on type 2 diabetes given too few patients in this subset (n=2). The analysis was restricted to unrelated individuals of European ancestry, leaving 91 carriers of *GDF15* PTVs (*Supplementary file 4j*, male N=42, female N=49, mean age = 56.3) and 40,000 randomly selected European-ancestry non-carriers (male N=20,000, female N=20,000, mean age = 56.9). In line with the MR analyses, we observed no differences in BMI (mean difference = 0.1, p-value = 0.90), WHR (male: mean difference = 0.01, p-value = 0.64, female: mean difference = 0.01, p-value = 0.55), glucose (mean difference = 0.14, p-value = 0.32), eBMD (mean difference = 0.02, p-value = 0.82), or HDL cholesterol (mean difference = 0.08, p-value = 0.08), suggesting that mono-allelic *GDF15* LOF does not have a strong effect on these traits. It is possible that the other allele is compensating for the LOF but given that no homozygous *GDF15* LOF was found in over 300,000 patients we were not able to test this hypothesis. It may be worthwhile for future analysis to experimentally explore the effect of heterozygous LOF on GDF15 protein levels.

## Discussion

In this study, we present a systematic phenotypic and genotypic analysis of GDF15 with a wide range of health outcomes and biomarkers. In line with previous findings, our analysis confirmed strong correlations of GDF15 plasma levels with a range of clinical parameters (e.g. age, smoking, and BMI) and human diseases (e.g. diabetes, cardiovascular and respiratory disease), as well as several inflammatory biomarkers. As preclinical data in mice strongly implicated Gdf15 in the aetiology of obesity and glucose tolerance, we specifically investigated whether human genetic evidence supports these findings. Using data from large biobanks, we found that neither MR nor *GDF15* PTV analyses supported a causal role for GDF15 plasma levels in influencing BMI, glucose, diabetes, or eBMD in humans. Instead, we found that higher BMI may cause increases in GDF15 plasma levels highlighting the role of GDF15 as a likely marker of metabolic stress in humans. Additionally, we replicated a causal association of GDF15 with HDL cholesterol and identified a nominal association with WHR.

Access to the FINRISK cohort provided a valuable opportunity to explore GDF15 plasma level associations with multiple phenotypes within a single large population. We found the strongest associations with all-cause mortality and cardiometabolic diseases (cardiovascular disease and diabetes), as well as cardiometabolic risk factors (e.g. hypertension, serum triglycerides, BMI), as previously reported (*Bonaca et al., 2011*; *Brown et al., 2002*; *Daniels et al., 2011*; *Ho et al., 2012*; *Kempf et al., 2007*; *Khan et al., 2009*; *Lind et al., 2009*; *Rohatgi et al., 2012*; *Wiklund et al., 2010*; *Wollert et al., 2007*). Our analyses also found that GDF15 plasma levels are a strong predictor of incident diabetes and an independent predictor of all-cause mortality, cardiovascular disease, and diabetes morbidity, suggesting its potential use as a pre-diabetic prognostic biomarker. We found associations with neoplasms of the lung and digestive system but not with other types of cancer. Our data further support the association of GDF15 plasma levels with inflammatory phenotypes and biomarkers (CRP, mid-regional pro-adrenomedullin: *Brown et al., 2007*) and uncovered less well-described associations with respiratory disease and psychiatric disorders. Despite previous observations suggesting a role for GDF15 in anorexia (*Borner et al., 2020a*), we did not identify any association of GDF15

plasma levels with this trait. However, we note that statistical power may have been limited due to the low number of cases (*Supplementary file 2c*). To identify phenotype associations in an unbiased way, we performed analyses encompassing all phenotypes available applying covariates uniformly across all analyses. We note that covariates specific to a particular disease might have been omitted introducing bias or leading to inflation or deflation of statistics. We detected a strong association between GDF15 levels and smoking, which was recorded and adjusted in our analyses as a binary phenotype. Smoking is an important contributing factor in multiple diseases and it is therefore likely that adjustment as a continuous variable would have been able to better explain the contribution of smoking to the phenotypic association signals, especially in respiratory disease, lung cancer, and some psychiatric disorders. These findings demonstrate that GDF15 plasma levels are a general marker for risk in multiple diseases and are associated with non-specific biomarkers such as CRP.

To define the genetic architecture of plasma GDF15 levels, we performed a GWAS meta-analysis in two large cohorts, FINRISK and INTERVAL, across three different assay platforms. A striking finding of this analysis was the substantial heterogeneity between variants across these studies, consistent with previous reports (*Jiang et al., 2018*). Exploring the LD between the variants displaying heterogeneity identified that a large proportion resided within an LD block encompassing a common missense variant, rs1058587 (p.H202D). This variant has been previously associated with hyperemesis gravidarum (*Fejzo et al., 2018b*) but the presence of this missense variant within this locus raises the possibility of epitope artefacts, indeed a previous study has identified epitope effects in this region (*Fairlie et al., 2001*). Therefore, it is highly likely that the inconsistency in GWAS results across this locus is driven by differences in the properties of the GDF15 assay used in each study (ILMA in FINRISK, SomaScan, and Olink assays in INTERVAL). Therefore, it may be beneficial to adopt a more consistent method for measuring GDF15 levels within the research environment. Approaches to assay protein levels that do not involve binding to epitopes, such as mass spectrometry, will be informative for avoiding potential artefacts due to protein-altering variants in the future.

Our analyses did not support a causal role for normal human GDF15 plasma levels with BMI, diabetes, glucose, or eBMD. These results were further supported by the lack of association between these phenotypes and *GDF15* PTV carriers in the UK Biobank dataset, albeit only heterozygous LOF carriers could be assessed. A meta-analysis completed by Cheung et al. (*Ho et al., 2012*) identified significant associations of genetically predicted GDF15 levels with HDL cholesterol (beta = −0.048, p-value = 0.001) and eBMD (beta = 0.026, p-value <0.001). Whilst we did not replicate the association with BMD, we do replicate the association of GDF15 with HDL. We additionally found a causal association of genetically determined GDF15 with WHR, although this did not survive multiple-testing correction in MR-PRESSO. This finding may be worth further study although the estimated effect is weak (beta = 0.017). We do not report a significant causal effect of genetically determined GDF15 with BMI consistent with a recent large-scale, pan-ancestry exome-sequencing study of over 640,000 individuals exploring the associations of rare coding variants with BMI that did not report an association with GDF15 (*Akbari et al., 2021*). Similarly a further study using genetic instruments weakly correlated to rs1058587 (strongest $R^2 \leq 0.02$) found no significant causal effect of GDF15 levels on BMI or diabetes traits (*Au Yeung et al., 2019*). In contrast to these results, *Karhunen et al., 2021* recently reported a significant causal association of GDF15 levels with BMI; the study used instruments that included SNPs in strong LD with rs1058587 the variant we identified to lead to a binding artefact in the assays, highlighting the importance of exploring assay bias in genetic studies. Additionally, the larger sample size of the study presented here (n=14,099) offers considerably improved statistical power over previous studies.

The absence of genetic data supporting a causal role for GDF15 plasma levels raises the possibility that GDF15 may behave differently in humans compared to rodents and non-human primates, where there are robust data demonstrating that GDF15 potently reduces body weight. GDF15 levels have also been demonstrated to provoke nausea-related pica and CTA responses in rat and mice, respectively (*Borner et al., 2020a*; *Patel et al., 2019*), as well as vomiting in shrews, *Suncus murinus*, but studies in non-human primates report no signs of nausea (*Xiong et al., 2017*). Several clinical trials testing the safety and efficacy of GDF15 therapy have been planned or initiated but no data have yet been disclosed. Nevertheless, body weight loss in pregnant women suffering from hyperemesis gravidarium has been linked with higher plasma GDF15 levels (*Fejzo et al., 2018a*). In FINRISK, information on nausea was not available but diseases with nausea as the main symptom were available

(such as gastroesophageal reflux disease) and no significant association with GDF15 plasma levels was found (OR = 1.1, p=0.63). In pregnancy GDF15 levels are found at much higher serum levels than normal (*Moore et al., 2000*) and it is therefore possible that at higher levels GDF15 may induce nausea and weight loss. Further studies into the effect of higher-concentration GDF15 in humans and the outcomes of clinical trials will elucidate its role further.

As our analyses did not support a causal role for normal human GDF15 plasma levels in the phenotypes examined, we sought to investigate if genetics supported the role for GDF15 as a consequence of these conditions. We conducted reverse MR using GDF15 plasma levels as the outcome and the previously assessed cardiometabolic traits and diseases as the exposures. We identified a significant causal association of BMI on GDF15 plasma levels, suggesting that higher GDF15 levels could be a consequence of higher BMI, supporting the recently reported role of GDF15 in response to stress (*Patel et al., 2019*). However, the contradictory findings of horizontal pleiotropy between MR-Egger and MR-PRESSO lead to residual uncertainty in the interpretation of the MR analysis. It is possible that the heterogeneity found in our study caused by the use of different assays is driving this pleiotropic finding. A recent report in a MR study of GDF15 levels performed by the SCALLOP consortium also found a significant causal effect of BMI (IVW effect = 0.20, p-value = $1 \times 10^{-7}$:*Folkersen et al., 2020*). Our replication of this finding strengthens the evidence that GDF15 levels are likely a consequence of differences in BMI. Further research will be required to determine conclusively the role of horizontal pleiotropy in this relationship.

Using valid genetic instruments in MR is of key importance as an invalid instrument would violate the assumptions of the model and lead to unreliable results. To minimise bias from differential assay binding, we conditioned on rs1058587 in our analyses, which could be overcautious. The association of rs1058587 with adiposity traits and hyperemesis gravidarum raises the possibility that GDF15 exerts a causal effect that is not mediated by altered plasma levels, such as via altered proteoforms and/or altered GFRAL binding. With the use of a wide range of GDF15 assays and the possibility that this missense variant is impacting the assay functionality in different assays to varying degrees, the generation of functional data quantifying assay performance is essential for understanding the contribution of this association to GDF15 plasma levels and improving the power of MR analyses. Similarly, as our *GDF15* PTV data is based solely on heterozygous carriers, it is possible that the other allele is compensating for the loss and further assessment of the impact of heterozygote *GDF15* PTVs on the presence of the protein will be required to validate our findings.

Taken together, our study provides a systematic and broad investigation of GDF15 phenotypic and genotypic associations, identifying possible epitope artefacts that could affect the data validity of GDF15 assays and introduce systematic bias in genetic analyses. Taking into account these biases, our genetic analyses did not support a causal association between normal human GDF15 plasma levels and obesity and diabetes, albeit a nominal finding of a causal association of WHR may warrant further investigation. Conversely, we found that increased GDF15 plasma levels may be a consequence of higher BMI, suggesting that GDF15 may act as a stress-induced biomarker. However, it is conceivable that at normal human plasma levels we are underpowered to detect GDF15's impact on BMI and elevated levels may induce a different or more pronounced effect. Additionally, whilst we have controlled for the assay bias caused by rs1058587, it is possible that this adjustment is too conservative. We caution that future studies with unbiased GDF15 measurements should reassess the impact of the variant using further Mendelian randomisation analyses. Our results do not support GDF15 plasma levels as a causal factor at normal human plasma levels for obesity or its related cardiometabolic diseases.

## Materials and methods
### Study population and phenotypes
#### FINRISK

This study was carried out in accordance with the recommendations of the Declaration of Helsinki. All participants of studies have given written informed consent. FINRISK study was approved by the Ethics Committee of Helsinki and Uusimaa Hospital District.

The FINRISK cohort comprises a cross-sectional population survey carried out over a 40-year period from the year 1972 across multiple regions in Finland. The study aimed to assess the risk

factors of chronic diseases and health behaviours in a working age population (*Borodulin et al., 2018*; *Vartiainen et al., 2000*) and consisted of 6000–8800 individuals per survey. Measurement of GDF15 plasma levels using an ILMA was included for participants from the 1997 recruitment cohort. Participants were additionally matched to their electronic health records giving access to longitudinal prescription records, death records, and diagnosis history. The cohort characteristics are summarised in *Supplementary file 1a*. In total, 6610 Finnish individuals from the FINRISK 1997 cohort with available GDF15 plasma concentrations and up to 676 disease outcomes were included in this analysis.

Nurse assessment/interview, self-report data, and blood samples were all collected at various time points during the study. A wealth of quantitative biomarkers (e.g. blood lipids, blood sugar markers, inflammatory biomarkers, cytokines, blood count, fatty acids, metabolome) and physiological measures (e.g. anthropometrics, cardiovascular physiology, body composition) were collected from the study participants. In addition, participants who consented were matched to their electronic health records giving access to hospital discharge registry (years 1969–2015), hospital discharge registry of specialist health care operations (1996–2015), death registry (1992–2015), drug reimbursement registry (1964–2015), drug purchase registry (1995–2015), and cancer registry (1953–2014). Gender was obtained from social security numbers. A group of clinicians working in the FinnGen consortium constructed 676 disease endpoints combining information from multiple health registries. ICD8, ICD9, and ICD10 codes were utilised in the disease endpoint definitions, for more information see https://www.finngen.fi/en/researchers/clinical-endpoints. Genotyping was carried out for in 6538 individuals who were recruited in 1997, for more information see Supplementary materials.

## INTERVAL

The INTERVAL study is a prospective cohort study comprising approximately 50,000 participants nested within a pragmatic randomised trial of blood donors (*Jammah, 2015*). Between 2012 and 2014, blood donors aged 18 years and older were recruited at 25 NHSBT (National Health Service Blood and Transplant) donor centres across England. Participants were generally healthy because people with a history of major diseases (such as myocardial infarction, stroke, cancer, HIV, and hepatitis B or C) and those who have had recent illness or infection were ineligible to donate blood. Participants completed an online questionnaire which included questions about demographic characteristics (e.g. age, sex, and ethnicity), anthropometry measures (height, weight), and lifestyle information (alcohol intake, smoking, physical activity, and diet).

Informed consent was obtained from all participants and the INTERVAL study was approved by the National Research Ethics Service (11/EE/0538).

For the SomaScan assays, two non-overlapping subcohorts of 2731 and 831 participants were randomly selected, of which 3301 participants (2481 and 820 in the two subcohorts) remained after genetic quality control.

For the Olink assay, protein measurements were conducted in an additional subcohort of 4998 INTERVAL participants aged over 50 at baseline; 4987 samples passed quality control for this panel and were included in the analyses.

For information on genotyping in the INTERVAl cohort, see Suplementary materials.

## UK Biobank

UK Biobank is a population-based cohort consisting of ~500,000 individuals with participants recruited between 2006 and 2010 and aged between 40 and 69. Recruitment and cohort information has been previously described (*Sudlow et al., 2015*). Electronic health records, self-report questionnaires, diet and lifestyle information, and biomarker data are available on this cohort. Here, we utilised BMI and WHR, which were measured during a nurse assessment, as well as diabetes information. Touchscreen questionnaire diabetic patients were applied in this study rather than ICD10 diagnosed in order to increase patient numbers, given the small subgroup of individuals being examined.

## Laboratory methods for GDF15 measurement
### FINRISK
Blood samples were collected after an advisory 4 hr fast, immediately centrifuged and then stored at –70°C until GDF15 measurement which was undertaken using an ILMA with a limit of detection of 24 ng/L and a linear range from 200 to 50,000 ng/L (*Kempf et al., 2007*). The ILMA is technically

identical to immunoradiometric assay (IRMA: *Horn et al., 1996*) except that the GDF15 detection anti-body was labelled with acridinium ester and assay results were quantified in a luminometer (Berthold). ILMA and IRMA use an antibody that binds over a sequence of Ala197-Ile308. GDF-15 concentrations measured with the ILMA and IRMA are very similar [(ILMA concentration/ILMA concentration)×100% = 97.8% ± 1.3%] and closely correlated ($r^2$=0.99, p<0.001, n=31 samples: H). Head-to-head comparison of IRMA with the clinical 'gold standard' Roche assay have been previously conducted and reported GDF15 levels were comparable (*Wollert et al., 2017*).

## INTERVAL

Blood sample collection procedures have been described previously (*Moore et al., 2014*). Blood samples were collected in 6 mL EDTA tubes and transferred at ambient temperature to UK Biocentre (Stockport, UK) for processing. Plasma was extracted by centrifugation into two 0.8 mL plasma aliquots and stored at –80°C before use.

The procedures for obtaining protein measurements using the SomaScan assay have also been described previously (*Sun et al., 2018*). Briefly, SomaScan is a multiplexed, aptamer-based approach that was used to measure the relative concentrations of 3622 plasma proteins or protein complexes using modified aptamers ('SOMAmer reagents', hereafter referred to as SOMAmers). Aliquots of 150 μL of plasma from INTERVAL baseline samples were sent on dry ice to SomaLogic Inc (Boulder, CO) for protein measurement. Modified single-stranded DNA SOMAmers were used to bind to specific protein targets and these are then quantified using a DNA microarray. Protein concentrations are quantified as relative fluorescent units. Quality control (QC) was completed at the sample and SOMAmer levels.

The Olink assay uses pairs of monoclonal antibodies (or a single polyclonal antibody) to bind each protein target, and then uses proximity extension followed by qPCR to quantify protein abundance. Aliquots of plasma from samples taken from INTERVAL participants at the 2-year follow-up survey were shipped on dry ice to Olink Proteomics (Uppsala, Sweden) for assay. Protein concentrations were recorded as normalised relative protein abundances ('NPX').

## Genotyping and imputation

### FINRISK

A total of 26,404 FINRISK samples were genotyped using several arrays: the HumanCoreExome BeadChip, the Human610-Quad BeadChip, the Affymetrix6.0, and the Infinium HumanOmniExpress (Illumina Inc, San Diego, CA). The present study, using samples taken in FINRISK in 1997, consisted of 6538 individuals which were genotyped using three genotyping arrays: the HumanCoreExome BeadChip, the Human610-Quad BeadChip, and the Infinium HumanOmniExpress (Illumina Inc, San Diego, CA). Genotype calls were generated together with other available datasets using zCall at the Institute for Molecular Medicine Finland (FIMM). After sample-wise QC (exclude samples with ambiguous gender, missingness [>5%], excess heterozygosity [±4 SD], non-European ancestry) and variant-wise QC (exclude SNPs with high missingness [>2%], low HWE p-value ($<1 \times 10^{-6}$), minor allele count (MAC) < 3 (in case Zcall'ed chip data) or MAC < 10 (chip data called using Illumina GenCall) steps, the samples were pre-phased using Eagle2 (version 2.3). Genotype imputation was carried out by using a Finnish population-specific reference panel consisting of 2690 high-coverage WGS and 5092 WES samples with IMPUTE2 (version 2.3.2: *Howie et al., 2009*) that allows the usage of two panels at the same time (the 'merge_ref_panels' option). Post-imputation QC involved excluding variants imputed with imputation INFO < 0.7.

### INTERVAL

The genotyping protocol and QC for the INTERVAL samples have previously been described in detail (*Astle et al., 2016*). DNA was extracted from buffy coat at LGC Genomics (UK) and was used to assay approximately 830,000 variants on the UK Biobank Affymetrix Axiom genotyping array at Affymetrix (Santa Clara, CA). Genotyping was performed in batches of approximately 4800 samples. Variants were excluded from a batch if they strongly deviated from HWE (p-value $< 5 \times 10^{-6}$) or had a within-batch call rate < 0.97. Sample QC included removing duplicate samples and samples with non-European ancestry, missing phenotypic sex and sex mismatches, and extreme heterozygosity

(±3 SD). Relatedness was removed by excluding one participant from each pair of close (first- or second-degree) relatives, defined as π > 0.187.

Additional variant QC steps were performed prior to imputation to establish a high-quality imputation scaffold. This included imposing a global HWE filter of p-value < 5 x 10⁻⁶, a call rate filter of 99% over the INTERVAL genotyping batches that a variant was not failed in, and a global call rate filter of 75% across all INTERVAL genotyping batches. All monomorphic variants, non-autosomal and multi-allelic variants were removed and 654,966 high-quality variants remained to be used for imputation. Phasing was conducted using SHAPEIT3 and variants were imputed using a combined 1000 Genomes Phase 3-UK10K imputation panel. Imputation was performed on the Sanger Imputation Server (https://imputation.sanger.ac.uk), resulting in 87,696,888 imputed variants.

## Genome-wide association analysis

### FINRISK

For the FINRISK cohort genome-wide association analyses were performed for 5817 individuals with plasma GDF15 concentrations available. A normal distribution of GDF15 plasma concentrations was achieved through applying an inverse normal transformation. Multi-dimensional scaling was done for genetic data of both studies using PLINK version 1.07 (*Purcell et al., 2007*). Only good quality autosomal markers passing the following criteria: imputation informativeness > 0.7, and MAF > 0.001, were included in further evaluations. Frequentist test with method 'expected', assuming an additive genetic model, was performed using SNPTEST version 2 (*Marchini et al., 2007*). Results were adjusted for the first five principal components of the genetic data to account for the population stratification and for gender, age, and genotyping set. The QQ and Manhattan plots and regional plots were created using R-2.11 to visualise genome-wide association results. The genomic positions indicated throughout this study are based on NCBI human genome build 37.

### INTERVAL

The GWAS for INTERVAL-SomaScan has previously been described in detail (*Sun et al., 2018*). Relative protein abundances were first natural log-transformed within each subcohort. Log-transformed protein levels were then adjusted in a linear regression for age, sex, duration between blood draw and processing (binary, ≤1 day/>1 day) and the first three principal components of ancestry from multi-dimensional scaling. A normal distribution of the protein residuals from this linear regression was achieved through rank-inverse normal transformation and these were used as phenotypes for association testing. Simple linear regression assuming an additive genetic model was used to test for genetic associations. Genotype uncertainty was accounted for by carrying out association tests on allelic dosages ('method expected' option) using SNPTEST version 2.5.2 (*Mullican et al., 2017*).

For the INTERVAL-Olink GWAS, normalised protein levels ('NPX') were first regressed on age, sex, plate, time from blood draw to processing (in days), and season (categorical: 'Spring', 'Summer', 'Autumn', 'Winter'). The residuals from this linear regression were rank-inverse normalised. The rank-inverse normalised residuals were then fitted in a linear regression model adjusted for ancestry by including the first three components of multi-dimensional scaling as covariates. GWAS was conducted using SNPTEST version 2.5.2.

### Fine mapping

Fine mapping was performed using the FINEMAP program (*Benner et al., 2016*). FINEMAP can potentially identify sets of variants with more evidence of being causal than those highlighted by a stepwise conditional analysis. The FINEMAP provides (1) a list of potential causal configurations together with their posterior probabilities and Bayes factors and, (2) for each variant, the posterior probability and Bayes factor of being causal. FINEMAP was applied with its default settings. LD between SNPs for the conditional analysis was combined using the formula: (N1 * LD1 + N2 * LD2 + N3 * LD3)/(N1 + N2 + N3), where N represents the number of individuals in the GWAS in each cohort, LD represents the $R^2$ value, and 1, 2, and 3 represent the three different cohorts: FINRISK, INTERVAL-SomaScan, and INTERVAL-Olink.

## Variant annotation

The functional significance of the fine mapped variants were explored using public databases and browsers, including *GTEx, Ensmbl, SNiPA, RegulomeDB* (*Võsa et al., 2021*).

## Meta-analysis

Meta-analysis was completed on the chromosome 19 locus around the GDF15 gene in METAL (*Willer et al., 2010*) using the IVW method. A total of 3,799 SNPs were found in common across FINRISK, INTERVAL-SomaScan, and INTERVAL-Olink.

## Statistical analysis

### Association study between GDF15 levels and disease endpoints and quantitative biomarkers

To examine the association of GDF15 with disease endpoints and quantitative biomarkers, logistic and linear regression models were used, respectively. Inverse variance transformed GDF15 levels were used in the analyses due to right-skewed distribution. Association of GDF15 with disease endpoints and quantitative biomarkers were examined in (1) age and sex, (2) age, sex, and BMI, (3) age, sex, and smoking-adjusted models. Analyses using linear and logistic regression models were performed using R2.11 (see URLs). FDR was applied to test for multiple test correction.

### Survival analysis

All analyses were performed in R version 3.4.0. Cox proportional hazard regression model was conducted to identify predictors of outcomes during 10 years. The survival curves for a Cox proportional hazards model were used to illustrate the timing of the death, type 2 diabetes, and CVD events during 10 years in relation to GDF15 quartiles, and statistical assessment between upper quartile and other quartiles was preformed using Cox proportional hazard regression model. To inspect the validity of the Cox model, the test of the proportional hazards assumption for a Cox regression model was used. Hosmer-Lemeshow goodness of fit test was used to determine whether models were calibrated with similar expected and observed event rates in both low- and high-risk individuals.

### Wilcoxon test for prevalent and incident disease

Wilcoxon rank sum test (Mann-Whitney U test) was performed, in R using function 'wilcox.test' from stats package, to test whether the distributions of the incident disease case group, prevalent disease case group, and control group were systematically different from one another. Wilcoxon rank sum test was used for the prevalent and incident analysis given non-transformed GDF15 plasma levels (that were not normally distributed) were used.

### Type 2 diabetes endpoint used for prevalent and incident analysis

The ICD codes used for type 2 diabetes endpoint definition in the incident prevalent analysis were: ICD9 codes: 2502A, 2501A, 2503A, 2504A, 2505A, 2506A, 2507A, 2508A, 2500A (diabetes mellitus, type 2); ICD10 codes: E110, E111, E112, E113, E114, E115, E11[6-8], E119 (adult type diabetes); ATC-code A10B (blood glucose lowering drugs, excluding insulins) from kela drug purchase register as well as ICD10 code E11 (adult type diabetes) from kela reimbursement register. During the ICD8 era, type 1 and type 2 diabetes were not separated in Finnish ICD8 diagnosis codes, and thus ICD8 diabetes diagnosis codes were not included to the type 2 diabetes endpoint. To ensure that type 1 diabetes cases are not mixed with type 2 diabetes cases, type 1 diabetes endpoint cases were removed from the type 2 diabetes endpoint cases.

## Mendelian randomisation

In brief, MR is a method that explores whether an intermediate trait has a causal relationship with an endpoint by using genetic instruments. The use of genetic data means there is less environmental bias. Variants identified as significantly associated with the intermediate trait are applied and summary results from intermediate trait – variants and endpoint – variants are applied in the model. MR-Egger regression does not constrain the intercept, therefore it is not biased by invalid IV but has reduced power. MR relies on three main assumptions that (1) the instruments are associated with the exposure,

(2) the instruments are not associated with any confounders, and (3) the instruments are associated with the outcome only through the exposure. Here, we applied random effect IVW analysis (which has the most statistical power but can be biased in the presence of horizontal pleiotropy), MR-Egger (which relies on the InSIDE assumption and has less statistical power but can be used to identify pleiotropy), weighted median MR (which is robust to outliers but sensitive to addition/removal of IVs), and MR-PRESSO (which is able to remove outliers but is prone to false positives in the presence of multiple invalid instruments). The strengths and weaknesses listed here are summarised in this paper (*Burgess et al., 2019*).

We applied MR (*Bowden et al., 2015*) to explore causality of GDF15 with BMI, WHR, glucose, type 2 diabetes, HDL cholesterol, and eBMD. We additionally explored these relationship in reverse: with GDF15 as an outcome and other traits as the exposure. GWAS summary statistics for GDF15 were taken from the conditional meta-analysis of FINRISK and INTERVAL for the forward MR and from FINRISK only for the reverse MR. GWAS in INTERVAL being completed on chromosome 19 only whereas GWAS in FINRISK were completed on all chromosomes. Summary statistics for the other traits were obtained from large public data resources: BMI (n=590,827: *Yengo et al., 2018*), WHR (n=485,486: *Shungin et al., 2015*), glucose (n=314,916: http://www.nealelab.is/uk-biobank), type 2 diabetes (n cases = 74,124, n controls = 824,006: *Mahajan et al., 2018*), HDL cholesterol (n=315,133: http://www.nealelab.is/uk-biobank), and eBMD (n=426,824: *Morris et al., 2019*). For BMI, MR with GDF15 as the exposure was applied using summary statistics from UK Biobank only (n=361,194) as the LD clumped variants were not available in the larger meta-analysis, that was utilised for reverse MR. We utilised LD clumping to identify independent genetic variants in summary statistics and applied these as genetic instruments in MR analyses. LD clumping was carried out in PLINK (version 1.9) with the following thresholds applied: $R^2 < 0.01$, p-value $< 5 \times 10^{-8}$ and clump radius < 3000 kb. MR analysis was run in R (version 4.0.2) using the package 'MendelianRandomization'. The IVW method from this package was applied to test for causality and a sensitivity analysis using MR-Egger to testing for horizontal pleiotropy. A finding of significant causality is indicated with the IVW p-value and a significant MR-Egger intercept p-value indicates a finding of horizontal pleiotropy. Weighted median MR and MR-PRESSO (Mendelian randomisation pleiotropy residual sum and outlier: *Verbanck et al., 2018*) were additionally applied as a further sensitivity analysis and to test for horizontal pleiotropy and was run in R using the 'MRPRESSO' package. The Global p-value indicates a significant finding of horizontal pleiotropy and the software also tests for outliers, the finding of which instigates removal of outliers from the analysis. MR-PRESSO then re-runs IVW testing and gives a significant distortion p-value to indicate if the removal of outliers reduced the horizontal pleiotropy found in the analysis.

## GDF15 PTV analysis
### UK Biobank exome sequencing
UK Biobank was whole exome sequenced (paired-end 75 bp) at Regeneron Pharmaceuticals using the IDT xGen version 1 capture kit and NovaSeq6000 for more information, see previous publication (*Wang et al., 2021b*). Data was available on 302,362 individuals of which >95% of CCDS had at least ×10 coverage and average coverage of CCDS was ×59. Alignment to GRCh38 and SNV and indel calling were completed utilising Illumina DRAGEN Bio-IT Platform Germline Pipeline version 3.07 on a custom-built Amazon Web Services (AWS) cloud platform. Annotation of SNVs and indels was performed using SnpEFF v4.3 against Ensembl Build 38.92 and MAPQ < 30 were excluded. Related individuals were identified using KING and first- and second-degree relatives were excluded from downstream analysis. Sex and ancestry checks were completed using PEDDY (*Pedersen and Quinlan, 2017*) and individuals with gender mismatch were excluded.

### Quality control
Individuals with *GDF15* PTVs were identified by extracting variants annotated as PTVs (including exon loss variants, frameshift variants, start lost, stop gained, stop lost, splice acceptor variants, splice donor variant, rare amino acid variant, transcript ablation, gene fusion, and bidirectional gene fusion). *Supplementary file 4j*, demonstrates the frequency of LOF variants in each ancestry. From the individuals that did not carry *GDF15* PTV variants 20,000 males and 20,000 females were randomly extracted as controls.

## Mann-Whitney U test

Analysis was completed in R (version 3.2.4) using 'wilcox.test' and models compared BMI, WHR (in males and females separately), diabetic status (self-reported), eBMD, and HDL in *GDF15* PTV carriers and non-carriers.

## Acknowledgements

We thank all study participants for their generous participation in FINRISK.

We thank the participants and investigators in the UK Biobank study who made this work possible (Resource Application Number 26041). We thank the UK Biobank Exome Sequencing Consortium (UKB-ESC) members AbbVie, Alnylam Pharmaceuticals, AstraZeneca, Biogen, Bristol-Myers Squibb, Pfizer, Regeneron, and Takeda for funding the generation of the data and Regeneron Genetics Center for completing the sequencing and initial QC of the exome sequencing data. We thank the AstraZeneca Centre for Genomics Research Analytics and Informatics team for processing and analysis of sequencing data.

Participants in the INTERVAL randomised controlled trial were recruited with the active collaboration of NHS Blood and Transplant England (https://www.nhsbt.nhs.uk/), which has supported field work and other elements of the trial. DNA extraction and genotyping was co-funded by the National Institute for Health Research (NIHR), the NIHR BioResource (http://bioresource.nihr.ac.uk), and the NIHR Cambridge Biomedical Research Centre (BRC-1215-20014) [*]. SomaLogic assays were funded by Merck and the NIHR Cambridge BRC (BRC-1215-20014) [*]. The academic coordinating centre for INTERVAL was supported by core funding from: NIHR Blood and Transplant Research Unit in Donor Health and Genomics (NIHR BTRU-2014-10024), UK Medical Research Council (MR/L003120/1), British Heart Foundation (SP/09/002; RG/13/13/30194; RG/18/13/33946), and the NIHR Cambridge BRC (BRC-1215-20014) [*]. A complete list of the investigators and contributors to the INTERVAL trial is provided in reference [**]. The academic coordinating centre would like to thank blood donor centre staff and blood donors for participating in the INTERVAL trial.

This work was supported by Health Data Research UK, which is funded by the UK Medical Research Council, Engineering and Physical Sciences Research Council, Economic and Social Research Council, Department of Health and Social Care (England), Chief Scientist Office of the Scottish Government Health and Social Care Directorates, Health and Social Care Research and Development Division (Welsh Government), Public Health Agency (Northern Ireland), British Heart Foundation and Wellcome.

This work was conducted as part of an alliance between the University of Cambridge and the AstraZeneca Centre for Genomics Research (AZ Ref: 10033507).

The Genotype-Tissue Expression (GTEx) Project was supported by the Common Fund of the Office of the Director of the National Institutes of Health, and by NCI, NHGRI, NHLBI, NIDA, NIMH, and NINDS. The data used for the analyses described in this manuscript were obtained from the GTEx Portal on 8 December 2021.

*The views expressed are those of the author(s) and not necessarily those of the NIHR or the Department of Health and Social Care.

**Di Angelantonio E, Thompson SG, Kaptoge SK, Moore C, Walker M, Armitage J, Ouwehand WH, Roberts DJ, Danesh J, INTERVAL Trial Group. Efficiency and safety of varying the frequency of whole blood donation (INTERVAL): a randomised trial of 45,000 donors. Lancet. 2017 Nov 25;390 (10110):2360–2371.

## Additional information

### Competing interests

Eleanor M Wigmore, Maria Fritsch, Ruth March, Dirk S Paul: are employees of AstraZeneca. Rachel MY Ong: is currently an employee of GlaxoSmithKline (although was not when this work was carried out). Veikko Salomaa: has received honoraria from Novo Nordisk and Sanofi for consulting. He also

has ongoing research collaboration with Bayer Ltd (all outside this work). Adam S Butterworth: reports grants outside of this work from Biogen, BioMarin, Bioverativ, Merck, Novartis, Pfizer and Sanofi and personal fees from Novartis. Athena Matakidou: is an employee of AstraZeneca and currently an employee of GlaxoSmithKline (although not an employee of GlaxoSmithKline when this work was carried out). The other authors declare that no competing interests exist.

### Funding

| Funder | Grant reference number | Author |
|---|---|---|
| NIHR Cambridge Biomedical Research Centre | BRC-1215-20014 | Rachel MY Ong |
| Sydäntutkimussäätiö | | Veikko Salomaa |

The funders had no role in study design, data collection and interpretation, or the decision to submit the work for publication.

### Author contributions

Susanna Lemmelä, Eleanor M Wigmore, Formal analysis, Investigation, Writing – original draft, Writing – review and editing; Christian Benner, Methodology, Writing – review and editing; Aki S Havulinna, Adam S Butterworth, Investigation, Methodology, Writing – review and editing; Rachel MY Ong, Formal analysis, Investigation, Writing – review and editing; Tibor Kempf, Kai C Wollert, Stefan Blankenberg, Tanja Zeller, Veikko Salomaa, Data curation, Writing – review and editing; James E Peters, Maria Fritsch, Ruth March, Writing – review and editing; Aarno Palotie, Mark Daly, Mervi Kinnunen, Supervision, Investigation, Writing – review and editing; Dirk S Paul, Athena Matakidou, Supervision, Investigation, Writing – original draft, Writing – review and editing

### Author ORCIDs

Susanna Lemmelä http://orcid.org/0000-0001-6027-8715
Eleanor M Wigmore http://orcid.org/0000-0003-0864-9990
Aki S Havulinna http://orcid.org/0000-0002-4787-8959
James E Peters http://orcid.org/0000-0002-9415-3440
Dirk S Paul http://orcid.org/0000-0002-8230-0116

### Ethics

Human subjects: FINRISK study was approved by the Ethics Committee of Helsinki and Uusimaa Hospital District. Informed consent was obtained from all participants and the INTERVAL study was approved by the National Research Ethics Service (11/EE/0538). All study participants provided informed consent and the UK Biobank has approval from the North-West Multi-centre Research Ethics Committee (MREC; 11/NW/0382).

### Decision letter and Author response

Decision letter https://doi.org/10.7554/eLife.76272.sa1
Author response https://doi.org/10.7554/eLife.76272.sa2

## Additional files

### Supplementary files

• Supplementary file 1. Characteristics of the FINRISK cohort. (**a**) FINRISK cohort characteristics. (**b**) Baseline characteristics in FINRISK associated with growth differentiation factor-15 (GDF15) plasma levels.

• Supplementary file 2. GDF15 plasma level associations and prognostic assessment. (**a**) Growth differentiation factor-15 (GDF15) level disease associations corrected for age and sex only. Abbreviations: SAH, aneurysmal subarachnoid haemorrhage; ANGIO, coronary angioplasty; CABG, coronary artery bypass grafting; DVT, deep vein thrombosis. (**b**) GDF15 level quantitative biomarker associations corrected for age and sex only. All quantitative biomarkers were rank-based inverse transformed. (**c**) GDF15 level disease associations corrected for age, sex, smoking, and BMI. Abbreviations: SAH, aneurysmal subarachnoid haemorrhage; ANGIO, coronary angioplasty; CABG, coronary artery bypass grafting; DVT, deep vein thrombosis. (**d**) GDF15 level associations

with prevalent and incident disease. Abbreviations: CNTRL, control. (**e**) Independent predictors of all-cause mortality, type 2 diabetes and cardiovascular disease (CHD or STR) event. Cox proportional hazard model was used to estimate the associations between risk factors and outcomes. (**f**) GDF15 level biomarker associations corrected for age, sex, smoking, and BMI. All quantitative biomarkers were rank-based inverse transformed. (**g**) GDF15 level disease associations corrected for rs1058587 as well as age, sex, smoking, and BMI. Abbreviations: SAH, aneurysmal subarachnoid haemorrhage; ANGIO, coronary angioplasty; CABG, coronary artery bypass grafting; DVT, deep vein thrombosis. (**h**) GDF15 level biomarker associations corrected for rs1058587 as well as age, sex, smoking, and BMI. All quantitative biomarkers were rank-based inverse transformed.

- Supplementary file 3. Genome-wide association and meta-analysis of the FINRISK and INTERVAL studies. (**a**) Significant (p-value < 5 × 10$^{-8}$) variants from genome-wide association study for FINRISK. (**b**) Significant (p-value < 5 × 10$^{-8}$) variants from genome-wide association study for INTERVAL-SomaScan. (**c**) Significant (p-value < 5 × 10$^{-8}$) variants from genome-wide association study for INTERVAL-Olink. (**d**) Meta-analysis of fine mapped GDF15 genome-wide association study variants in FINRISK and INTERVAL. Variants listed in table were identified by fine mapping genome-wide association study (GWAS) results from FINRISK and INTERVAL. LD blocks were defined as SNPs that had LD > 0.1 with the lead variant (most significantly associated variant). (**e**) Linkage disequilibrium (R$^2$) between fine mapped variants in FINRISK, INTERVAL-SomaScan, and INTERVAL-Olink. (f) Significant (p-value < 5 × 10$^{-8}$) variants from genome-wide association study for FINRISK conditioned on rs1058587. (**g**) Significant (p-value < 5 × 10$^{-8}$) variants from genome-wide association study for INTERVAL-SomaScan conditioned on rs1058587. (h) Significant (p-value < 5 × 10$^{-8}$) variants from meta-analysis of genome-wide association study for FINRISK, INTERVAL-SomaScan, and INTERVAL-Olink. (**i**) Functional annotation of GDF15 meta-analysis fine mapped variants. Canonical transcripts only are shown. Results were obtained from Ensembl variant effect predictor. Frequencies were obtained from 1000 genomes.

- Supplementary file 4. Results from Mendelian randomisation assessment.
 (**a**) Estimates and F-statistics for instrumental variables used in Mendelian randomisation (MR) assessment with growth differentiation factor-15 (GDF15) as the exposure (forward MR). (**b**) Weighted median Mendelian randomisation results for forward MR of GDF15 with cardiometabolic traits. (**c**) Estimates and F-statistics for instrumental variables used in Mendelian randomisation of body mass index (BMI) with GDF15 as the outcome (reverse MR). (**d**) Estimates and F-statistics for instrumental variables used in Mendelian randomisation of waist-hip ratio (WHR) with GDF15 as the outcome (reverse MR). (**e**) Estimates and F-statistics for instrumental variables used in Mendelian randomisation of diabetes with GDF15 as the outcome (reverse MR). (**f**) Estimates and F-statistics for instrumental variables used in Mendelian randomisation of glucose with GDF15 as the outcome (reverse MR). (**g**) Estimates and F-statistics for instrumental variables used in Mendelian randomisation of high-density lipoprotein (HDL) with GDF15 as the outcome (reverse MR). (**h**) Estimates and F-statistics for instrumental variables used in Mendelian randomisation of estimated bone mineral density (eBMD) with GDF15 as the outcome (reverse MR). (**i**) Weighted median Mendelian randomisation results for reverse MR of GDF15 with cardiometabolic traits. (**j**) GDF15 protein-truncating variant carrier frequency in UKB.

- Transparent reporting form

### Data availability

Participant-level genotype and phenotype data from UK Biobank are available by application: https://www.ukbiobank.ac.uk/enable-your-research/apply-for-access. Participant-level genotype and phenotype data (as part of the FinnGen consortium) are available by application: https://www.finngen.fi/en/access_results. INTERVAL-SomaScan participant-level genotype and protein data, and full summary association results from the genetic analysis are available through the European Genotype Archive (accession number EGA00001002555). Summary association results are also publically available at http://www.phpc.cam.ac.uk/ceu/proteins/, through PhenoScanner (http://www.phenoscanner.medschl.cam.ac.uk) and from the NHGRI-EBI GWAS Catalog (https://www.ebi.ac.uk/gwas/downloads/summary-statistics). INTERVAL-Olink summary association results are publically available at http://www.phpc.cam.ac.uk/ceu/proteins/.

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
