## [Editor Report]

This integrated observational and genetic analysis using data from large biobanks comprehensively investigated the role of Growth Differentiation Factor-15 in a wide range of human diseases and will be of interest to cardiometabolic disorder researchers. GDF-15 appears to be a marker of metabolic stress rather than having a causative role.

---

## [Decision Letter]

**Decision letter after peer review:**

Thank you for submitting your article "Integrated Analyses of Growth Differentiation Factor-15 Concentration and Cardiometabolic Diseases in Humans" for consideration by *eLife*. Your article has been reviewed by 3 peer reviewers, including Edward D Janus as Reviewing Editor and Reviewer #1, and the evaluation has been overseen by Matthias Barton as the Senior Editor.

Essential revisions:

1) There is room for improvement concerning Mendelian randomization. Whilst the use of Mendelian randomization design may circumvent issues with confounding in observational studies, there was a lack of certain analyses which may improve the robustness of findings, as well as concerns over the instruments which appeared to be correlated.

The study is in line with a previous Mendelian randomisation study exploring the relation of GDF-15 in type 2 diabetes (https://pubmed.ncbi.nlm.nih.gov/31161347/) although different from another MR study (https://pubmed.ncbi.nlm.nih.gov/33686698/). It would be useful to include these studies in the Introduction, and compare and contrast the findings in this paper with previous MRs in the Discussion.

2) Please briefly explain how MRs may be less vulnerable to confounding and the underlying assumptions needed. Please also mention the assumptions of the IVW and sensitivity analyses. Please consider the use of multiplicative random effect model for IVW which is more robust to outlier influence. Would also be good to include weighted median method. R2 for LD should be more stringent (i.e. <0.01 or <0.001). Otherwise, the analyses used would be invalid given the use of correlated variants. The authors could also consider using analyses which take into account correlated variants. Please also provide more information in how the genetic instruments were identified (e.g. p value threshold). There is also no mention of instrument strength such as F statistics.

3) Why were meta analyses only performed for SNPs around the GDF-15 gene instead of the entire sets of SNPs across different chromosomes? (Page 26)

4) Given the authors mentioned metformin it would be worth discussing this paper (https://pubmed.ncbi.nlm.nih.gov/27974345/) although metformin use is associated with higher GDF-15.

5) Please provide more details regarding which phenotypes were used to define type 2 diabetes. I presume these included the use of self reports and anti-diabetic medications?

6) In the PTV analyses there were only 2 type 2 diabetes cases so is such analysis valid.

7) Please provide further elaboration on the rationale of the Willcoxon test for prevalent and incident diabetes.

8) Why was GDF-15 transformed? I think this is not necessary for using GDF-15 as a predictor in a regression and would complicate the interoperation of the unit change.

9) For the association analysis of GDF15 plasma levels with disease outcomes, symmetrically grouping relevant disease outcomes may be more informative and this will eliminate the redundancy of the same subject being included in multiple outcomes.

10) For the GDF15 plasma concentration associations with prevalent and incident cases, why was only diabetes was analyzed? At least pulmonary related diseases, as a whole, should be analyzed. This section is very informative.

11) More analysis should be done for the GDF15 and biomarker associations instead of simple descriptive analysis.

12) If BMI is the driving force for GDF15, how to explain that GDF15 has much stronger effects than BMI in the Cox proportional hazard analysis presented in Figure 1?

13) More detailed analysis with BMI could help to close the loop of BMI-GDF15-mortality.

14) The truncated variants had no significant effects but given the subjects also had a normal allele this presumably compensated for the truncated allele. A homozygote or compound heterozygote for a non functioning allele could be highly informative but will probably be rare and hard to find. This could be further discussed.

---

## [Author Response]

Essential revisions:1) There is room for improvement concerning Mendelian randomization. Whilst the use of Mendelian randomization design may circumvent issues with confounding in observational studies, there was a lack of certain analyses which may improve the robustness of findings, as well as concerns over the instruments which appeared to be correlated.The study is in line with a previous Mendelian randomisation study exploring the relation of GDF-15 in type 2 diabetes (https://pubmed.ncbi.nlm.nih.gov/31161347/) although different from another MR study (https://pubmed.ncbi.nlm.nih.gov/33686698/). It would be useful to include these studies in the Introduction, and compare and contrast the findings in this paper with previous MRs in the Discussion.

Thank you for drawing our attention to these studies. We have now included them in our manuscript in both the introduction:

“Opposing results have also been reported regarding the association of GDF15 on BMI, with one study finding a causal relationship (Karhunen, Larsson, and Gill, 2021) and another reporting no significant effect (Au Yeung, Luo, and Schooling, 2019)”.

And discussion:

“Similarly a further study using genetic instruments weakly correlated to rs1058587 (strongest R^2^≤0.02) found no significant causal effect of GDF15 levels on BMI or diabetes traits (Au Yeung et al., 2019). In contrast to these results, Karhunen et al., recently reported a significant causal association of GDF15 levels with BMI; the study used instruments that included SNPs in strong LD with rs1058587 the variant we identified to lead to a binding artefact in the assays (Karhunen et al., 2021), highlighting the importance of exploring assay bias in genetic studies”.

We have commented on the robustness of our MR analyses and selection of instruments below.

2) Please briefly explain how MRs may be less vulnerable to confounding and the underlying assumptions needed. Please also mention the assumptions of the IVW and sensitivity analyses. Please consider the use of multiplicative random effect model for IVW which is more robust to outlier influence. Would also be good to include weighted median method. R2 for LD should be more stringent (i.e. <0.01 or <0.001). Otherwise, the analyses used would be invalid given the use of correlated variants. The authors could also consider using analyses which take into account correlated variants. Please also provide more information in how the genetic instruments were identified (e.g. p value threshold). There is also no mention of instrument strength such as F statistics.

Thank you for your suggestions. We have now implemented a random-effect IVW analysis and reduced the R2 threshold to < 0.01 (R2 < 0.001 resulted in too few genetic instruments). The updated results can be found in Tables 3 and 4 and in the text on pages 13-16. Briefly, the horizontal pleiotropy is reduced across all MR analyses and we additionally find significant associations with HDL as well as WHR (although this does not survive multiple-testing correction in MR-PRESSO). We have added the results for the weighted median analysis in the supplementary materials (Supplementary File 4, Tables 4b and 4i).

We have also added more information on the assumptions of the methods applied as well as the description of how the genetic instruments were identified and added the F-statistics. To ensure strong instruments were applied, we set a p-value threshold of 5x10^-8^.

The assumptions of the MR techniques are discussed in the methods section and read:

“Mendelian randomisation relies on three main assumptions (1) that the instruments are associated with the exposure, (2) the instruments are not associated with any confounders and (3) the instruments are associated with the outcome only through the exposure. Here we applied random-effect IVW analysis (which has the most statistical power but can be biased in the presence of horizontal pleiotropy), MR-Egger (which relies on the InSIDE assumption and has less statistical power but can be used to identify pleiotropy), Weighted Median MR (which is robust to outliers but sensitive to addition/removal of instrumental variables) and MR-PRESSO (which is able to remove outliers but is prone to false positives in the presence of multiple invalid instruments). The strengths and weaknesses listed here are summarised in this paper”.

We have added the following to the manuscript to provide further information on how the instruments were identified: “LD clumping was carried out in PLINK (version 1.9) with the following thresholds applied: R^2^<0.01, p-value < 5x10-8 and clump radius < 3000kb”.

F-statistics are reported for all exposure SNPs in Supplementary File 4, Tables S4a and S4c-g.

3) Why were meta analyses only performed for SNPs around the GDF-15 gene instead of the entire sets of SNPs across different chromosomes? (Page 26)

We have updated the Results section to read: “We then performed a meta-analysis across the genome for all three cohorts. Only signals within 1Mb of the GDF15 gene reached significance in the meta-anlaysis.”. Additionally we have updated Figure 2 to include the full GWAS Manhattan plot.

4) Given the authors mentioned metformin it would be worth discussing this paper (https://pubmed.ncbi.nlm.nih.gov/27974345/) although metformin use is associated with higher GDF-15.

Thank you for drawing our attention to this interesting paper. We have commented on this in the introduction to expand our discussion on the relationship between GDF15 with Metformin. It now reads: “In fact, an analysis of changes induced by Metformin treatment in 237 blood biomarkers showed GDF15 levels to be the most significantly altered, independent of glucose and glycosylated haemoglobin”.

5) Please provide more details regarding which phenotypes were used to define type 2 diabetes. I presume these included the use of self reports and anti-diabetic medications?

We thank the reviewer for noticing this point. The FINRISK analyzes in the manuscript were performed during the FinnGen piloting project. Both ICD9/10 codes from the Hilmo inpatient, outpatient registers and cause of death registers; ATC-codes from KELA drug purchase register; and ICD10 codes from KELA drug reimbursement register were used for defining these endpoints. The endpoint used in T2D prevalent/incident analysis is E4_DM2 (E4 Type 2 diabetes), from which E4_DM1 (E4 Type 1 diabetes) endpoint cases have been removed to ensure that there are no type 1 diabetes cases in the T2D case group. More details can be found here https://www.finngen.fi/en/researchers/clinical-endpoints. These disease endpoints have been reviewed by clinical groups within FinnGen.

We have now included a full definition in the methods section:

“The ICD-codes used for Type 2 diabetes (T2D) endpoint definition in the incident prevalent analysis were: ICD9 codes: 2502A, 2501A, 2503A, 2504A, 2505A, 2506A, 2507A, 2508A, 2500A (diabetes mellitus, type 2); ICD10 codes: E110, E111, E112, E113, E114, E115, E11[6-8], E119 (adult type diabetes); ATC-code A10B (blood glucose lowering drugs, excluding insulins) from KELA drug purchase register as well as ICD10 code E11 (adult type diabetes) from KELA reimbursement register. During the ICD8 era, type1 and type2 diabetes were not separated in Finnish ICD8 diagnosis codes, and thus ICD8 diabetes diagnosis codes were not included to the T2D endpoint. To ensure that Type 1 diabetes cases are not mixed with Type 2 diabetes cases, T1D endpoint cases were removed from the T2D endpoint cases.”

6) In the PTV analyses there were only 2 type 2 diabetes cases so is such analysis valid.

Thank you for raising this point. We have removed this analysis and rephrased this within the text to now read:

“We were not able to carry out this analysis on type 2 diabetes given too few patients in this subset (n=2)”.

7) Please provide further elaboration on the rationale of the Willcoxon test for prevalent and incident diabetes.

We thank the reviewers fot this suggestion and have expanded the rationale on the use of the Wilcoxon test below.

Wilcoxon rank-sum test is a nonparametric test used to compare two independent samples, while Wilcoxon signed-rank test is used to compare two dependent samples. The Wilcoxon rank-sum test is commonly used for the comparison of two groups of nonparametric, not normally distributed, data. Non-parametric tests have two advantages over parametric tests: they do not require the assumption of normality of distributions and they can deal with outliers. A Student’s t-test for instance is only applicable if the data are Gaussian or if the sample size is large enough (usually n≥30, thanks to the central limit theorem). A non-parametric test should be used in other cases.

In prevalent-incident T2D (and other endpoints) analysis non-transformed GDF15 values were used, which were not normally distributed. In addition, the sample size of prevalent T2D was only 37. Thus we decided to use Wilcoxon rank sum test in T2D prevalent/incident analyses.

We have now specified the test used and additionally have added the sentence*:*

*“*Wilcoxon rank sum test was used for the prevalent and incident analysis given non-transformed GDF15 plasma levels (that were not normally distributed) were used.*”*

8) Why was GDF-15 transformed? I think this is not necessary for using GDF-15 as a predictor in a regression and would complicate the interoperation of the unit change.

We thank the reviewer for raising this point. The GDF15 distribution was not normally distributed and had large tails/outliers. Given these outliers may represent potential participants of interest, we took the decision not to remove them and inverse normalise the data. Whilst we understand this is not always necessary for a dependent variable in regression, we wanted to eliminate any potential bias that could be introduced by skewed data.

9) For the association analysis of GDF15 plasma levels with disease outcomes, symmetrically grouping relevant disease outcomes may be more informative and this will eliminate the redundancy of the same subject being included in multiple outcomes.

Thank you for this suggestion. We have tried to be as inclusive as we can in our analysis and have therefore considered as many phenotypes as possible, but we acknowledge that there may be redundancy. The argument for ‘clumping’ or ‘splitting’ is an ongoing discussion that has not yet come to a definitive conclusion (e.g. see https://www.nature.com/articles/s41593-020-0609-7); here, we have included both the parent groups and the subgroups. These disease phenotypes have been reviewed by a clinical team within FinnGen. We have applied FDR multiple testing correction to allow for overlapping phenotypes but agree that there is a chance some nominal results are of interest. However, given this more stringent threshold of significance we do have confidence in the disease and biomarker associations that we report.

10) For the GDF15 plasma concentration associations with prevalent and incident cases, why was only diabetes was analyzed? At least pulmonary related diseases, as a whole, should be analyzed. This section is very informative.

We thank the reviewer for noticing this point. As suggested, we have now explored GDF15 plasma concentrations in prevalent and incident cases in the top associated disease endpoints; type 2 diabetes, chronic obstructive pulmonary disease; atherosclerosis (excluding cerebral, coronary and PAD); psychiatric disorders and malignant neoplasm of respiratory system and intrathoracic organs. We have added results to the manuscript and Supplementary File, Table S2d.

11) More analysis should be done for the GDF15 and biomarker associations instead of simple descriptive analysis.

As advised by the editor (correspondence on 27^th^ May 2022), this comment was not addressed.

12) If BMI is the driving force for GDF15, how to explain that GDF15 has much stronger effects than BMI in the Cox proportional hazard analysis presented in Figure 1?

Thank you for raising this interesting question. Whilst we have presented evidence that BMI may be causal for changes in GDF15, it is also possible that this is not specific to BMI. Obesity (measured using BMI) was a more significant predictor of diabetes mellitus but not cardiovascular disease. GDF15 is upregulated in many diseases and is also associated with CRP, indicating it may well be a non-specific biomarker. This is also demonstrated in its effect on all-cause mortality in Figure 1(a).

13) More detailed analysis with BMI could help to close the loop of BMI-GDF15-mortality.

As advised by the editor (correspondence on 27^th^ May 2022), this comment was not addressed.

14) The truncated variants had no significant effects but given the subjects also had a normal allele this presumably compensated for the truncated allele. A homozygote or compound heterozygote for a non functioning allele could be highly informative but will probably be rare and hard to find. This could be further discussed.

Thank you for this comment. We agree and have now included in the Results section of the revised manuscript:

“It is possible that the other allele is compensating for the LOF but given that no homozygous *GDF15* LOF was found in over 300,000 patients we were not able to test this hypothesis. It may be worthwhile for future analysis to experimentally explore the effect of heterozygous LOF on GDF15 protein levels”.

We have additionally added the following to the discussion “… albeit only heterozygous LOF could be assessed”.